# A Design Concept for a Tourism Recommender System for Regional Development

**Leyla Gamidullaeva** [1,*] **, Alexey Finogeev** [2]**, Mikhail Kataev** [3] **and Larisa Bulysheva** [4]

1  Department of Marketing, Commerce and Service, Penza State University, Krasnaya st., 40, Penza 440026, Russia
2  CAD Department, Penza State University, Krasnaya st., 40, Penza 440026, Russia
3  Department of Radioelectronic Technologies and Environmental Monitoring, Tomsk State University of Control Systems and Radioelectronics, Tomsk 634034, Russia
4  Information Technology and Decision Sciences Department, Old Dominion University, Norfolk, VA 23529, USA
*  Correspondence: gamidullaeva@gmail.com

**Abstract:** Despite of tourism infrastructure and software, the development of tourism is hampered due to the lack of information support, which encapsulates various aspects of travel implementation. This paper highlights a demand for integrating various approaches and methods to develop a universal tourism information recommender system when building individual tourist routes. The study objective is proposing a concept of a universal information recommender system for building a personalized tourist route. The developed design concept for such a system involves a procedure for data collection and preparation for tourism product synthesis; a methodology for tourism product formation according to user preferences; the main stages of this methodology implementation. To collect and store information from real travelers, this paper proposes to use elements of blockchain technology in order to ensure information security. A model that specifies the key elements of a tourist route planning process is presented. This article can serve as a reference and knowledge base for digital business system analysts, system designers, and digital tourism business implementers for better digital business system design and implementation in the tourism sector.

**Keywords:** tourism industry; regional development; tourism product; tourist route; information recommender system; blockchain; digital avatar; traveling salesman problem





## 1. Introduction

Tourism is one of the most noticeable elements of international trade in the non-commodity sector of the economy. According to the World Tourism Organization, it is the third-largest export category in the world export, being the top export product for most developed countries. Tourism is one of the industries having the highest multiplier effect on the economy. Investments in the tourism industry form the added value in transport, trade and services, construction and production of building materials, and other types of economic activity. An important socio-economic effect of tourism development on the population involved in the creation and provision of services is the growth of employment and incomes, and the formation of entrepreneurial culture [1].

The tourism industry has undergone dramatic changes since various forms of information and communication technologies (ICT) began to penetrate the society, industry, and markets. Modern world trends in the tourism industry are related to the modernization and development of user interfaces along with the digitalization of promotion tools; creating communities in social media and services having information and educational content; integration of domestic tourism products into communication (navigation and cartographic) services, and voice assistants; promotion of tourism products using strategies based on technologies for studying and predicting consumer preferences, etc. [2].

The Tourism Development Strategy of the Russian Federation until 2035 [1] highlights such major digital solutions as introduction and development of big data technologies and artificial intelligence (AI) for data collection and analysis; development of a system for the promotion of tourist services; formation of the best travel offers, taking into account tourist wishes, weather conditions, traffic situation, etc.; development of online services for tourist route planning along with buying tickets and booking hotels. Applying big data analytics (behavioral models, information on current events and world news, tourist preferences, meteorological data, etc.) allows building tourist routes, analyzing their economic efficiency, etc.

Tourism development requires an appropriate infrastructure and, accordingly, software to figure out common travel challenges. It should be noted that, despite of tourism infrastructure and software, the development of tourism is hampered due to the lack of information support, which encapsulates various aspects of travel implementation.

It is advisable that an application should contain necessary preferable tourist information on movement, hotels, routes, attractions, etc. The application should find out the trip features and recommend a tourist the best options for activities.

The current ICT development features the Internet as a means of communication between the global population and various organizational structures (airlines, hotels, shops, etc.). There are different types of tourist traveling (individual, mass, and specialized) to be taken into account. Each of these can be pedestrian, automobile, water and consider tourist health status, financial capabilities, etc. In such a variety of choices, special features of tourist routes, considering semi-structured (sometimes conflicting) travel requests, should be kept in view. In this regard, travel companies should actively participate in the development of route recognition and competitiveness with the world's leading tourist routes.

Thus, the authors can conclude that it is necessary to develop a universal software and a recommender system to promote design and implementation of individual tourist routes. The present study is aimed at solving this issue.

A recommender system (RS) in tourism is an intelligent computer system that provides valuable suggestions and serves as a tool for planning a tourist trip [3–5]. Basically, a RS is a mechanism that filters information for the end users they are interested in. Personalized data search and discovery applications that help users identify and select useful items and information are at the core of the recommender process [6]. These systems display most relevant information to the end users based on their personal profiles provided initially, taking into account the behavior of other similar users. In this way, a RS is like a filter that adjusts the information available on the Internet regarding the interests and preferences of the end users.

The complexity of developing such software is due to a variety of formal models that describe movement, accommodation, routes, attractions, taking into account time, cost, sequence of visits, etc. Any tourist should be able to consider expert assessments necessary for thinking through and clearly defining each route element.

In our opinion, when developing a RS in the field of travel and tourism, it is essential to consider the recommendation context, a group of users to receive a joint benefit from the recommended elements, and temporal relationship between the latter.

Furthermore, traditional recommender systems satisfy the needs of tourists only by studying a limited number of factors. However, there are numerous determinants, such as environmental factors, actual geographic coordinates, travel destinations, user preferences, etc. to be taken into account when building individual travel routes in order to provide travelers with a reliable recommendation. The existing recommender systems offer tourists to choose the most suitable transport, accommodation, destinations, places of special interest, and other attributes necessary for the trip. However, it has always been about the introduction of information technologies into individual business processes of tourism and hospitality industry enterprises in the arrangement of particular implementation stages of the tourist route. It should be recognized that such business processes as booking sys-

tems, e-commerce, mobile applications, image recognition systems, big data processing, etc. have already been rebuilt in the sphere of tourism and hospitality industry having a significant impact on the competitiveness of tourist destinations and other tourism market entities [7–12]. It is noteworthy that modern technologies allow integrating business processes at various levels of economic system functioning [12–14]. When building individual tourist routes, a synthesis of various approaches and methods is required to develop a universal tourism information recommender system.

This paper is an advancement of research works on the development of recommender systems and fundamentals of business process management [8,9]. The study objective is proposing a concept of a universal information recommender system for building a personalized tourist route. The concept includes an original authorial DLT-based methodology for tourism product synthesis in order to consistently form a digital tourist avatar being a cyberphysical system. The end result of the recommendation service is the synthesis of alternative TPs with the recommendation of the optimal route, followed by navigation support for the traveler to adjust the route, taking into account the current situation, associated costs, user wishes, as well as to save notes and reviews of the traveler about points on the route for accumulating experience, analysis, and synthesis of other tourism products. Each product synthesized and mastered during the journey is stored in a hierarchical database synthesized in the form of a distributed ledger (blockchain) and hashed for unique identification in order to protect against unauthorized access and compromise. This is necessary because in modern travel services, booking services, ratings, etc. predominantly positive reviews are written by managers of tourist facilities, hired bloggers, or bots. At the same time, negative reviews and descriptions are deleted since access to them is determined by system administrators of information services of tourist sites. That is why, in order to collect and store information from real travelers, this paper proposes using elements of blockchain technology to ensure information security.

This work is organized as follows: In Section 2, related works on recommender system in the tourism context are presented. The section concludes with a gap analysis and an explanation of how our work addresses these noted gaps. Then, our research conceptual framework is described in Section 3, including problem statement and content criteria used as parameters of a tourism product. Data collection and preparation for tourism product synthesis is also presented here. In Section 4, the authors develop a methodology for tourism product formation. Drawing on these results, the authors further propose a blockchain-based tourism product representation. Finally, the results are discussed in Section 5 to highlight the implications, opportunities for future research, recommendations and limitations of our work.

## 2. Literature Review

There are numerous studies devoted to the development of recommender systems in tourism employed in various domains, suggesting different types of items involving activities that are experienced by movies, restaurants, travel destinations, etc. [10,11]. Researchers provide different approaches: hybrid recommendation approach [12], approach based on collaboration and text analysis [13], agent-based personalized approach [14], etc. However, various AI techniques are currently used to solve the problem of reliability for tourist recommendations [15,16]. Bedi, et al. propose using reputation-based collaborative filtering algorithm that augments reputation to existing collaborative approach for generating relevant recommendation in tourism domain.

Intelligent agents are able to analyze user behavior, learn user profile automatically, and provide proactive recommendations depending on the current context [14].

Some systems go beyond the list of recommended tourist attractions and use automatic planners to schedule recommendations within a route that can last several days [17,18].

Other approaches take into account the opening and closing times of attractions, or the time required to get from one point to another, and offer a detailed schedule and route plan based on that. However, such planning is very complicated with unguaranteed optimal

solution. Some systems solve this problem using such AI-based optimization techniques as ant colony and metaheuristic approach, or iterative methods [19].

Baltrunas et al. [20] outline a recommender system for tourist points of interest (POI) considering 14 contextual factors that account for the travel time, weather conditions, the user mood, or the type of group that accompanies the traveler. In this case, each factor is endowed with a finite set of possible values. Massimo and Ricci [21] present a novel approach to recommendations that helps tourists to choose key POI. The authors consider situations when there are few users (POI visitors) and no additional information about users, other than their past experience of visited POI, is available. User behavior is first modeled by grouping users with identical visit trajectories and then by learning a common user behavior pattern similar for everyone. In the research [22], instead of using POI, the authors propose to leverage the POI category to better investigate the user's interests. They offer a point-of-interest (POI) category recommendation model based on group preferences and indicate that the PPCM has improved recommendation performance compared to other models.

Automatic clustering algorithms can be used to categorize tourists with similar preferences or characteristics [23].

Methods of approximate reasoning such as fuzzy logic or Bayesian networks can be used to drive fuzzy data associated with user preference inference [24–27]. Liu, et al. [28] argue that the recurrent neural network-based approaches ignore the high-order relationship between users and POIs. They propose a novel approach for successive POI recommendation with an improved graph convolution network for learning the dynamic representation of users and POIs.

An example of work in the field of application of industrial information integration methods for solving transport problems is the article [29], which considers the problem of optimizing vehicle routes in order to reduce energy consumption and environmental pollution. A model for multi-purpose optimization of multiple routes using an elitist genetic sorting algorithm without dominance is proposed to minimize the cost and time spent in the process of vehicle movement, as well as to reduce carbon emissions. The final route is selected in the decision-making process according to the method of gray relational analysis and based on the assessment of the value of information entropy. The authors of another article [30] propose a two-stage hybrid algorithm for solving the problem of vehicle routing in order to optimize logistics and planning. The paper [31] proposes a multicriteria model for optimizing movement between a set of points to minimize the total number of routes, travel costs, and reduce the number of long routes. A multi-purpose optimal model of vehicle routes with minimum total cost, carbon emissions, and a minimum probability of an accident risk, taking into account environmental factors, is also proposed in [32].

Most IoT systems transmit data streams from sensor nodes through routers to cloud servers, which cause problems with the confidentiality and integrity of information when transmitted through public communication channels, processed on cloud servers of third-party providers, etc. Blockchain technologies allow solving these problems.

Here is the list of existing software products used in tourism.

The YouRoute [33] program contains places, route maps, descriptions and photos of attractions, information about hotels, car rentals, and flights. The program users can save the routes in their personal accounts and recommend them in the public domain. Despite numerous advantages of the program, there are some drawbacks associated with the inability to set the method of movement along the route (on foot, plane, car, etc.) to provide a detailed description of the route. The SygicMaps [34] program is a web-based mapping service that presents a selection of attractions, hotels, and travel routes, and describes the route traveled. The ClassTravel [35] application presents the possibilities to buy tickets for various types of transport, or to book hotels, and gives a detailed description of shopping spots. The Maps.Me [36] application provides offline maps to plan a route around cities with indicated streets, house numbers, transport, shops, attractions, etc.

As a rule, these software programs and applications offer visual description of various places and show the route traveled or planned. Some applications provide more advanced visualization of tourist POI, and the Krpano [37] software for showing all kinds of panoramic and virtual 3D images of premises and POI on the web as an example. The Spherika [38] program allows creating virtual tours and combining spherical panoramas with various transition functions.

Unfortunately, all applications used for travel planning have limitations concerning functionality of the route builder: building a route either comes down to connecting POI with straight lines on the map, or the builder does not take into account all means of transportation and the user personal preferences.

There are several web mapping applications on the market: Yandex Maps [39]; 2GIS [40]; Google Maps [41]; Wikimapia [42]. All maps are approximately the same in terms of functionality and allow showing the city, POI, transport, or building a route. But they sometimes are unable to provide feedback in the form of images that are geographically linked to certain places.

Considering the individual characteristics and capabilities of tourists, as well as vaguely formulated, vague or conflicting requests of different tourists in groups leads to a large variety and number of TPs, especially when it is necessary to take into account the different requests of members of the tourist group, which should also be taken into account when modeling business processes. The most well and fully developed in this regard are the models of tourism business processes that are used in travel agencies and tour operators to develop the attractiveness of TP for travelers and increase their competitiveness. The business process models of users who prefer to independently plan and select their own TP without contacting specialists are often implemented based on their own experience or the experience of other tourists, information about which is retrieved from the Internet and is often unverified, subjective, or false.

Thus, the creation of a universal application free from the identified shortcomings is an urgent task.

## 3. Conceptual Framework

Designing a tourism product requires a comprehension of its key parameters. In this regard, a tourism product is understood as a detailed tourist route formed by an intelligent system in an interactive mode based on data mining and a matrix for traveler preferences, data for the synthesis of new tourism products, and data on previously formed products of other tourists.

In particular, the following criteria are used as parameters of a tourism product:

**(a) Type of tourism product** (recreational; medical; business; sports; gastronomic; wildlife; wine; beach; ethnographic; religious; pilgrimage; transit; educational; children's; military–patriotic);

**(b) Mode of transportation (transport; walking):**

- Type of transport: personal transport (car, motorcycle, bicycle, electric transport, etc.), public transport (air, rail, road, water-motor ships, cruise ships, ferries, yachts, boats);

**(c) Traveler characteristics:**

- Identification data (passport, birth certificate, residential address and registration, telephone, e-mail);
- Financial data (bank card, account);
- Personal data (age, weight, gender, presence of physical limitations);
- Data on accompanying persons (number of adults and children, identification, and personal data);

**(d) Preferred tourist route characteristics:**

- Travel territory (country, region, city, locality);
- Spatial and temporal characteristics of the trip (place of departure and arrival, start/end time, number of stops (overnights) on the route, etc.);

- Approximate number of tourist sites (key points of the route selected on the map) to be visited for the entire time and per day;
- Average time of visiting a tourist site (depending on the choice of the type of excursion service or without it): up to 30 min; 30–60 min; 60–90 min; 90 min and over;
- Attraction fees for one visit;
- Remoteness of tourist sites from places to stop (overnight stays) depending on the type of product (e.g., for city excursions—-a distance from the center, from the tourist (historical) center, pedestrian streets, museums, attractions, etc.; for out-of-town excursions—-a distance from stops (roads) to POI (for backpackers and tourists with transport); for beach tourism—-a distance from water resources; for gastronomic and wine tourism—-a distance from exclusive dining destinations, etc.);

**(e) Characteristics of accommodation places (stops, overnight stays):**

- Type of accommodation: city inn (hotel); resort hotel; sanatorium; rest house; recreation center; boarding house; apart-hotel; motel; hostel; country hotel; tourist base, recreation center [43]; apartment; flat; guest house; room; house; cottage; villa;
- Cost of living (a single person or a room) per day;
- Availability of basic services (parking, Internet, kitchen, shops and cafes within walking distance, number of beds, toilet, and bathroom, etc.);

**(f) Preferred food characteristics:**

- Type of preferred dining destination (restaurants, bars, canteens, cafes, snack bars) [44];
- Type of meals at the place of residence (breakfast, half board, full board, all inclusive, ultra all inclusive, self-catering);
- Cost of one meal check (up to 500 rubles, up to 1000 rubles, up to 1500 rubles, etc. or average food expenditures per day) (Figure 1).

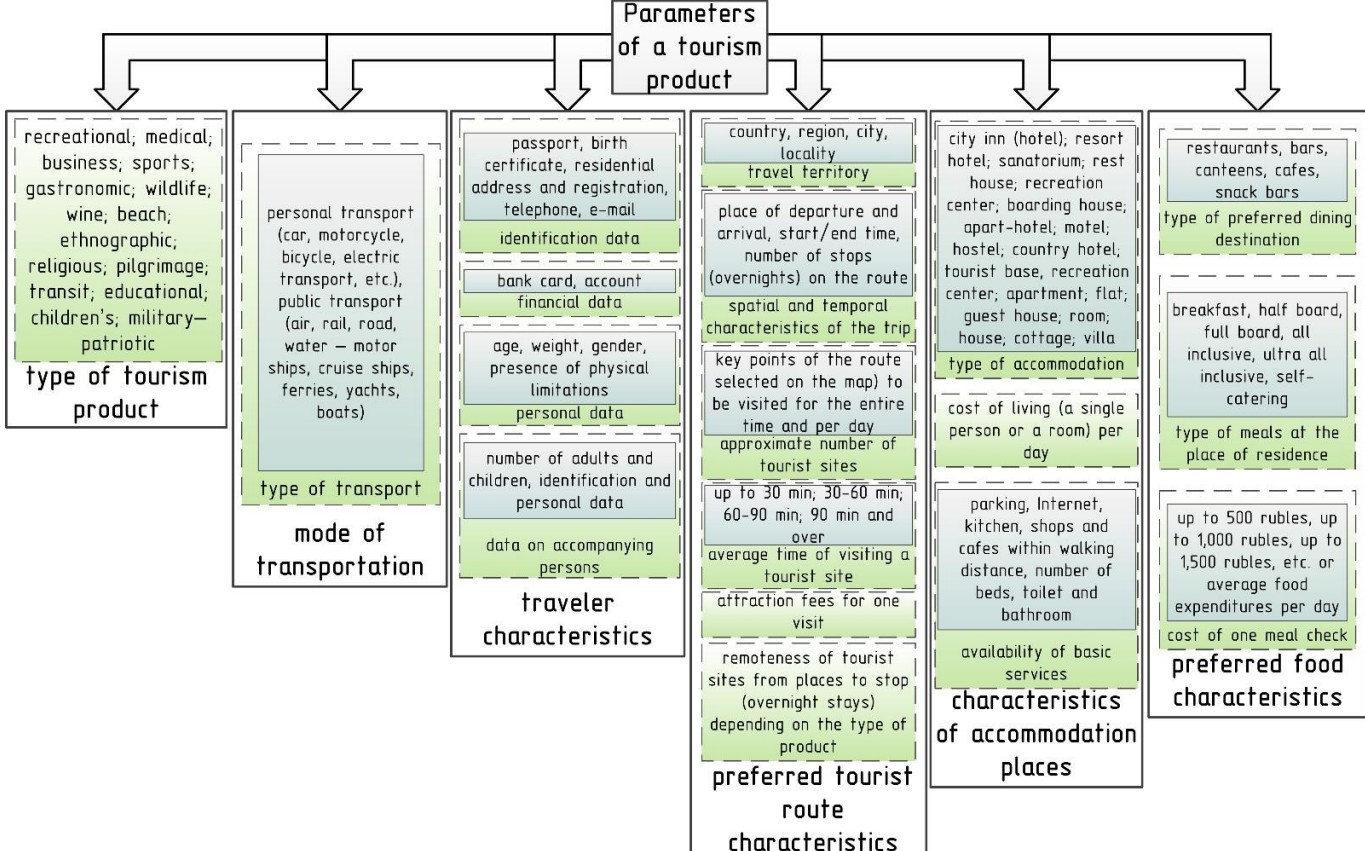

**Figure 1.** Parameters of a tourism product.

*Data Collection and Preparation for Tourism Product Synthesis*

A synthesized detailed tourism product incorporates start/end time and place of the trip; a mapped route track indicating numerous key points (attractions, places of stops, parking, overnight stays, meals, excursions, etc.) with geospatial tags; time lags between the key points of the movement track; cost parameters of movement (transport tickets, fuel) and visits to key points (entry, excursion service); tourist information (personal, identification, financial, etc.). In particular, personal data of travelers, such as age and weight, are necessary to determine the length of the route and the average speed of movement, taking into account physical restrictions when visiting certain places. Spatial data and time lags are needed to specify movement logistics, select the type of transport, purchase tickets, etc. due to faraway attractions.

First and foremost, detailed information about the key points of the selected or synthesized tourist route (sights, places of stops, overnight stays, meals, gas stations, etc.) is required for the users. Therefore, a major issue is to integrate the developed system with well-known services, and to collect and consolidate open data sources on the Internet. Since the existing systems do not provide direct access to the collected data, integration therewith is performed through links for using third-party services (booking, financial settlements, movement monitoring, ticket purchases, etc.). It should be noted that the use of third-party services may cause additional financial costs.

However, there is a lot of free data on the key places of the tourist route to be obtained from open sources on the Internet. Tourists also need real travel reviews providing latest (within several months) information about the route and key points when forming or choosing a tourism product. Therefore, another challenge is to search for and collect up-to-date information from different sources with its initial consolidation in the repository, subsequent monthly updating, and deletion of obsolete data. It is especially important to update information about the state of roads, the quality of service at places of stops, overnight stays and meals, the state and fees of visiting attractions, prices for excursions, etc.

Generally, information about the key points of the tourist route is presented in a variety of sources; therefore, it is necessary to remove duplicates and consolidate data in the process of collecting thereof. While accumulating traveler reviews, greater focus should be paid on negative reviews containing much more useful information. Moreover, too many positive reviews without comments lack true and necessary information as they are often written by interested parties.

To consolidate and check for duplicates, information about waypoints retrieved by crawlers is converted into keyword vector models [45,46]. The implemented synthesis of vector models is based on the modified Word2Vec algorithm with the calculation of the sampled logistic loss function in the process of predicting and optimizing the learning model [47]. Geospatial labels are used to compare vector representations of point descriptions. To compare information by the degree of updating, timestamps of extracted descriptions on source sites are used.

## 4. Results

### 4.1. Methodology for Tourism Product Formation Based on User Preferences

The methodology for the formation and recommendation of tourism products to end users is presented in the form of a number of major stages, which, in turn, are a sequence of algorithmic steps implemented in the form of client and server components of a digital tourism recommender ecosystem. The first two stages are the collection of data on the key points of possible tourism products and the collection of data on the tourism preferences of users, including their personal data. Both steps are performed in parallel by different applications (Figure 2).

| Collection, cleaning and consolidation of big data on the possible components of future tourism products | | Collection, cleaning and consolidation of the personal information of the tourist and data about his preferences and opportunities. | | Synthesis and clustering of tourism products. |

**Figure 2.** Generalized algorithm for developing an individual tourism product.

At the same time, the information received earlier should be periodically updated to update the data, both about key points and about the user and his changing capabilities and preferences.

(A) The first stage is the collection, cleaning, and consolidation of big data on the possible components of future tourism products.

The components of a detailed tourism product are:

(a) Spatial and temporal parameters of the start and end points of routes. Retrospective and predicted weather data specific to the spatial and temporal parameters of the routes.

(b) Sets of key points of the future route, including the coordinates of tourist attractions, the coordinates of places of possible stops (parking lots), the coordinates of public catering places, the coordinates of places for boarding public transport (airports, railway stations and bus stations, places for ordering transfers, car rentals, carsharing, etc.), coordinates of places of accommodation for tourists, tourist and information centers, etc.

(c) Spatial parameters and time lags of movements between key points. Spatial and temporal data are needed to determine the logistics of travel, select the type of transport, purchase tickets, etc. since attractions can be located at remote distances.

(d) Cost parameters of movement (cost of tickets, fuel, transport rental, transfer), cost parameters of visits to key points (cost of entry, excursion service, parking, etc.).

(e) Sufficiently detailed information about the key points of the tourist product (attractions, places of stops, overnight stays, meals, gas stations, etc.). The data are collected by various mapping and navigation services (Google Map, Maps.Me, Yandex Map, etc.), as well as booking services (Booking, AirBrib, Ostrovok, Kvartirka, Sutochno.ru, etc.), which are in the public domain access. Therefore, an important task is the integration of the recommender ecosystem with third-party services to obtain the required information by users, as well as the availability of links to various sites associated with key points on the Internet. Since existing services do not provide the ability to download the collected data, the integration is the addition of many links to third-party resources tied to the coordinates of key points.

(f) Links to reviews of travelers who have previously visited key points. Real reviews of key waypoints are important when synthesizing or selecting a tourism product according to user preferences. It is very important to have up-to-date feedback from other people (within a few months), including negative information.

1. Information about the key points of the tourist route and other parameters of the tourist route, as a rule, is presented in a variety of sources on the Internet, therefore this stage is implemented using the crawling technology with the cleaning of the received data from duplicates and information noise. The search and collection of information about key points is carried out by crawling various sources and consolidating links to resources with national information. At the same time, bypassing previously found links should be periodically repeated to update information and remove obsolete and empty links.

2. To reduce information noise at the consolidation stage, the found links to the same resources and duplicates of information on different sites and services are checked and excluded. The first task is solved quite simply by simply comparing links, and to

solve the second task, it is necessary to extract and compare textual descriptions. To do this, the extracted information about the key points is converted into vector word models using the modified Word2Vec algorithm with the calculation of the sampled logistic loss function to optimize the learning model in the comparative analysis of texts. To identify updated information, timestamps of adding data to source sites are used.

(B) The second stage is the collection, cleaning, and consolidation of the personal information of the tourist and data about his preferences and opportunities.

The main difficulty of working with a recommender ecosystem is determined by the need to model the process of choosing tourist products with a predictive assessment of time, cost, personal and other parameters of routes, taking into account various factors, such as route complexity, weather conditions in a given time frame, cost changes on the route, possible accidents, and incidents. The complexity increases significantly when synthesizing group routes, since one also has to consider the preferences, capabilities, and constraints from individual users in groups using a game theory model, such as the Nash equilibrium model. To take into account the various criteria for the optimality of routes and temporal, spatial, financial, and other factors, it is necessary to use multi-criteria optimization models. Moreover, the user should be able to make changes to the recommended tourist product after getting acquainted with the updated information about the key points of the route and feedback from other users about them.

1.  At the first step of this stage, users are asked to enter their personal data (personal, identification, financial, etc.) during the registration process in the recommender ecosystem. In addition to identification (passport and other data) and financial data (linking cards to pay for services), personal data also include age, weight, physical fitness assessment and some medical indicators that are necessary to recommend the complexity and length of routes, forecast average speed of movement, taking into account restrictions when visiting key points. The completeness of the information entered at this step further affects the optimization of the choice of a tourist product for a particular tourist and the life cycle management of the synthesized tourist product. Personal and personal data are virtual characteristics of a tourist's digital avatar and are stored in the profile, which is presented as a block of a distributed registry, the result of calculating the hash function as a unique identifier. Personal and personal data change when they are updated. With each change, the profile hash identifier is recalculated and re-indexed to combine all data blocks (blocks of previously selected products, blocks of tourist preferences) associated with his digital avatar. The characteristics of the avatar are further applied to cluster tourists according to their physical, financial, and other capabilities, linking to a set of possible tourist routes for this cluster.

2.  In the next step, the tourism preferences of the user are determined for the time interval chosen by him and the given geospatial wishes. To help users, a questionnaire with a digital bot is built according to the architecture of the recommender ecosystem to answer frequently asked questions and provide expert assistance to tourists when filling out questionnaire forms. A registered user must fill out a questionnaire every time their travel preferences change. The entered data about the user's travel preferences in a specific time interval is stored in the form of a distributed registry block and is associated with the hash identifier of his digital travel avatar, which is previously calculated from personal and personal data. If the preferences of the tourist do not change during the next selection of the route, then he does not need to fill out the questionnaire again and the previously formed block is used to select the product. When tourist preferences change, after each new filling of the questionnaire, a new block of the distributed registry is formed, which is also identified by a hash identifier of the digital avatar. If, in the process of working with the system, the user uses one set of preferences each time or rarely changes it, it is determined by a static avatar, the set of blocks of which practically does not change. A user who often changes his

preferences is considered a dynamic avatar and the set of blocks that characterize his avatar constantly grows. Blocks with information about tourist preferences of avatars are associated with a hash identifier, on the one hand, with blocks of personal and personal information, and, on the other hand, with tourist products synthesized at the next stage, the descriptions of which are also presented in the form of a chain of distributed registry blocks.

3.   In the next step, the digital avatars of tourists are clustered according to the criteria generated from personal data. Each cluster brings together users who are similar in personal, physical, financial, and other characteristics, which they entered in the first step. This step is necessary to group users in order to offer them selecting only those tourism products that meet medical, physical, financial, and other restrictions. When changing personal and personal data, clustering for each user is performed again after entering new data.

4.   The next step of this stage is to solve the problem of clustering digital avatars of tourists according to the characteristics of their tourist preferences, which they entered through the questionnaire. This step for static avatars is most often performed only once or rarely, in case of a change in his preferences. For a dynamic avatar, the clustering algorithm is implemented after each new input of travel preference data. The result of clustering is grouping into clusters similar in preferences of tourists. At the same time, the parameters of their preferences are averaged and assigned to virtual digital avatars of clusters, which are considered to be their centers and relative to which the distances for individual digital tourist avatars are calculated. The average parameters of the cluster avatar are recorded as a data block, for the identification of which the hash of the cluster identifier is calculated. When adding new preferences by a tourist, his digital avatar can be cloned and the clone falls into several clusters at the same time, if the new set of preferences is very different in distance from the center of the old cluster. If the set of preferences does not change too much and after recalculating the distances it is closest to the center of the old cluster, then cloning does not occur. In the future, the synthesis of tourist routes is performed according to a set of averaged preferences corresponding to the digital avatar of the cluster. The recommender system will offer all synthesized routes for a cluster avatar to all its tourist avatars. After selecting a tourist product from those offered by the system, its parameters and components are added to the personal tourist data block, which is identified by the hash identifier of the tourist avatar. If none of the options suits the user, then the user will be asked to synthesize their own tourist route based on the key points. The new route will be linked to his digital avatar with a double ID hash that includes his avatar's personal hash and the hash ID of the cluster to which the clone belongs.

(C) The third stage is the synthesis and clustering of tourism products.

One of the objectives of the study is the synthesis and selection of TP in accordance with the individual preferences of tourists, considering the acquired and third-party tourism experience, and the influence of external factors to optimize the life cycle of TP at all stages in order to rationally use the tourist and recreational resources of the regions of the Russian Federation. Each synthesized version of the tourist route represents a scheme of its movement between key points with temporal, spatial, financial, and other characteristics of the sites. The scheme is visualized on the map as a route of key points and sections of the route, and parameters, descriptions, and links to various information (text data, graphics, photos, and video materials), reviews of key points and sections are visualized through a drop-down menu when selected.

Tourist route synthesis can be performed in automated and manual modes. In an automated mode, a synthesis of a set of alternative routes is performed according to the matrix of tourist preferences of the digital avatar of the cluster, which the user got into at the previous stage, with quantitative estimates of the parameters of a possible tourist product. Tourist preferences can be represented not only by quantitative, but also by

qualitative parameters. To represent qualitative parameters, fuzzy linguistic variables are used. Therefore, in the general case, the complete preference matrix in the initial state has a fuzzy nature.

1. The first step of the stage of automated synthesis is the solution of the fuzzification problem for the transition to a purely quantitative matrix. During the procedure, specific values of membership functions of fuzzy terms are determined based on the initial data, which represent the set $V = \{v_1, v_2, \dots, v_m\}$. The parameter $v_i \in X_i$, where $X_i$ is the universe of the linguistic variable $\beta i$ for which the set of conditions of the form $< \beta_i$ is considered refers to $\alpha j >$ with the membership function $\mu_v^{(K)}$ n for the variable $\beta_i$ Since the value of vi is used as an argument, a set of quantitative values is thus found, which are the result of the fuzzification of conditions.

2. Next, to identify the preference matrix, a hash of its contents is calculate d, which is then used as a unique identifier of a tourist product synthesized for a specific user. As a geospatial reference of the route, the spatial coordinates of the places of its beginning and end are used. For visual identification of the tourist product selected by a specific user, he is assigned a picture with a stylized image of the tourist's avatar. If a route is chosen by a group of tourists, then a set of avatar images that have chosen it at a given time is assigned to it. The number of avatars for each route determines the statistics of its choice and its popularity. This statistic allows users to choose routes by popularity or, for example, choose routes where there are a small number of tourists at a given time. For a set of alternative routes that have not yet been selected by specific users, an image generated by the system is installed to identify the digital avatar of the cluster.

3. The spectrum of possible tourist routes is synthesized according to the preference matrix of the avatar of the centroid of the cluster to which the user belongs. Automatic synthesis of possible routes is implemented by an algorithm based on the traveling salesman method with an estimate of time lags and approximate costs for travel, accommodation, meals, sightseeing, excursion services, etc. If such routes have been synthesized earlier, then they are simply offered to the user to choose from. Each user in the cluster, after setting the start and end points of the route, travel time interval, selection of attractions and cost indicators, is offered a number of existing alternative routes associated with the cluster avatar. The hashes of the selected tourism products are added to the tourist's personalized travel profile block and further form a matrix of links to the traveled routes to evaluate his travel experience and offer similar products to him in the future. He can independently choose the product that suits him, let the recommender system bot choose the most optimal route according to the criteria he has chosen, or switch to the manual synthesis mode of his own unique product (point 4), which will later be associated with the user avatar and the cluster avatar using the double hashing method. In the second case, the multicriteria optimization module is activated to select the most suitable route, and the user is offered a number of optimization criteria. Optimization criteria are selected depending on the mode of travel of the tourist. Examples of sets of optimization criteria can be the following: (a) For hikers, the criteria can be: minimum distance between successive points of the route, maximum points on the route, taking into account the average speed of the tourist, taking into account the time for their inspection, the presence of stops, meals, toilets on the route; (b) for cyclists, the criteria can be: the minimum distance between successive points of the route, the maximum points on the route, taking into account the average speed of the cyclist, taking into account the time for their inspection, the availability of stops, food, toilets and overnight on the route; (c) for car tourists traveling by public transport, the criteria can be: minimum travel time (taking into account the travel time between points in accordance with the traffic schedule) for a given travel time per day, minimum travel costs or a given ticket price, maximum attractions with taking into account the time for their inspection, the availability of stops on the route, meals and overnight stays; (d) for tourists traveling by private

vehicle, the criteria can be: minimum travel time (including travel time between points) for a given travel time per day, minimum travel cost, maximum attractions during daylight hours, taking into account the time to see them, availability of gas stations, stops, meals, etc. on the route. In any case, for a tourist, the choice of the optimal route is related to the purpose of the trip, which can be determined by visiting a certain number of specific attractions, choosing the distance to travel between given points, financial criteria, etc. The conditions for solving the problem are difficult to find a single option, especially for a group of tourists. One of the options for solving the route selection problem is the traveling salesman method, when the initial conditions are written in the form of a matrix, where the rows correspond to the key points of the route, and the columns correspond to the criteria for selecting points. The most difficult to choose the best routes are cities with a lot of attractions that a tourist is going to see in a limited time, since possible routes include streets that intersect with each other, house numbers, various modes of transport, many places to sleep and eat with similar characteristics. Therefore, here there is the greatest number of options, the choice of which is a significant difficulty. The simplest solution to the problem is the "brute force" method, when all possible route options are considered to select the optimal one.

4. The formation of a tourist product in manual mode is implemented by the user on a digital map by setting key points of the route with a choice of interesting sights, places of residence, meals, stops, transport, etc. The set of parameters and descriptions of the components of the synthesized tourist product is recorded in the tourist product block and is associated by double hashing with the hash of the tourist avatar and the avatar of his cluster. A hash of the synthesized route is also calculated for its unique identification and binding to blocks with data on route components (key points). Thus, in a distributed registry, a tourist product is represented as a chain of blocks with a description of the route, its key points, personal and personal data of the tourist who chose it, a set of tourist preferences of the avatar of this tourist and a set of tourist preferences of the cluster avatar.

5. In the last step of this stage, the synthesized tourist products are clustered according to the degree of similarity of routes, descriptions of key points, and sites with similar characteristics in accordance with the preference matrices of digital tourist avatars. Clustering results are used to identify connections between routes created by different users, comparative analysis, and evaluation of alternative tourism products in order to select the best routes for a particular tourist and/or group according to preferences, features, and capabilities, as well as the likely integration of the most similar of them. Such tourist products will be called convergent. The degree of convergence determines the assessment of the similarity of tourism products (Figure 3).

*4.2. Blockchain-Based Tourism Product Representation*

The distributed ledger technology (DLT) is a way to ensure reliability, uniqueness, and integrity of digital files, and to enhance data resilience against hacks and cyberattacks [48]. To identify the data blocks in a distributed ledger, a unique identifier being the resulting hash value is used. It is added both to its header and to any transactions with block data. The ledger itself is formed from blocks of transactions linked by the previous block hash included into the data structures of the current block with calculating the hash thereof. In our case, operations to form and change data blocks with user information and tourist preferences in the distributed ledger of their digital avatars, and those with synthesized tourist routes in the distributed ledger of tourism products are termed as transactions. Thus, two chains of data blocks are formed, with the links between thereof being established when selecting tourism products and matching their parameters with tourist preferences. Blockchain data of tourism products contain characteristics of tourist preferences; data on digital user avatars; temporal, spatial, financial, and other logistics data about the tourist route; information about key points with text and graphics (photos, images, animation)

materials; audio and video recordings; links to third-party services; information resources for the user convenience when choosing or synthesizing a tourism product.

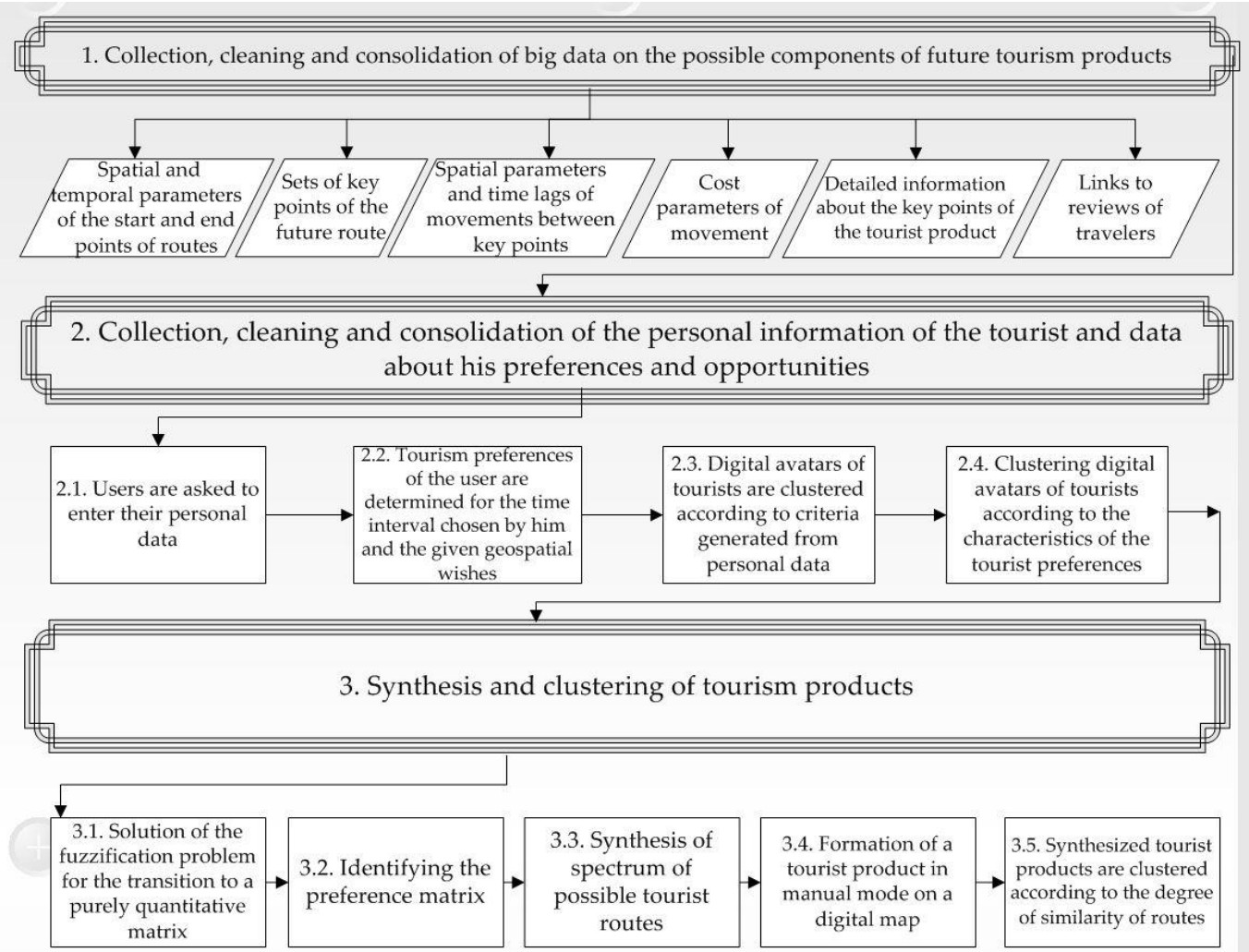

**Figure 3.** Stages of the algorithm for developing an individual tourism product.

The integrity of the accumulated information in distributed ledgers is ensured by triple hashing. The links of successive entries of each ledger are implemented by two hashes that are calculated both for the main entry and the metadata. Wherein, each calculated hashes is used to calculate another one, and the resulting hash is inserted into the header of the next entry. The third hash pair establishes a link between the two ledgers. When choosing a route for a particular user, the calculated digital avatar hash is inserted into the header of the data block with the selected tourism product, and, conversely, the hash of the selected product is added to the digital avatar data block. Next, the number of tourism product hashes accumulated in the user digital avatar block is used as a tourist experience estimate to be used in tourism rating depending on the number of trips. If the hash is not found in the entry or does not match after being calculated, then the entry is modified.

To authenticate user digital avatars and those of tourism products, a smart contract that executes the linearly homomorphic signature (LHS) algorithm of an avatar (product) [44,45] is used. The verified identifier of a digital avatar or product (unique signature) is synthesized by combining two hash functions:

(a) $H_1$ *(Id,(X,Y))*, where *Id* (digital avatar (product) identifier) and *X,Y* (latitude and longitude coordinates of the user location (starting point of the route) are the input data;

(b) $H_2$ *(T(x,y))*, where *T* is a point on the elliptic curve over a finite field with *(x,y)* coordinates:

$$\{(x,y) \in (\mathbb{R}_p)^2 \mid y^2 = x^3 + ax + b \ (mod \ p), 4a^3 + 27b^2 \neq 0 \ (mod \ p)\} \cup \{0\} \qquad (1)$$

where $\mathbb{R}_p$ is a finite field of $a$, $b$ integers modulo $p$, and 0 is a point at infinity.

The elliptic curve point is to enhance the security of digital avatar (product) data in order to complicate the procedure for potential signature decryption based on the elliptic curve discrete logarithm problem. The generated signature can be represented as a combined $S$ hash:

$$S = (H_1(Id,(X,Y)), H_2(T(x,y))) \qquad (2)$$

To calculate the first part $H_1(Id,(X,Y)$ of the digital signature, the secure hash algorithm with a digest size of 256 bits (SHA-256) is used, being applied as a bitcoin hashing algorithm in the blockchain. To synthesize the second part $H_2(T(x,y)$ of the digital signature, the elliptic curve digital signature algorithm (ECDSA) is used, being defined in the group of points on the elliptic curve.

A digital signature of an avatar (product) is generated according to the following algorithm (Figure 4):

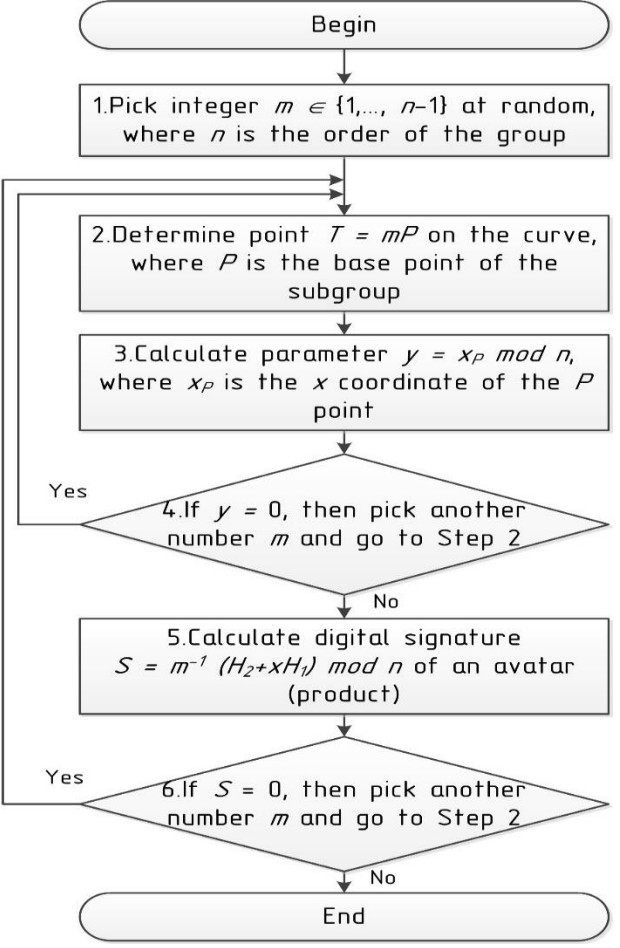

**Figure 4.** Algorithm for generation of an avatar (product).

Step 1: Pick integer $m \in \{1, \dots, n-1\}$ at random, where $n$ is the order of the group,
Step 2: Determine point $T = mP$ on the curve, where $p$ is the base point of the subgroup,
Step 3: Calculate parameter $y = x_P \ mod \ n$, where $x_P$ is the $x$ coordinate of the $p$ point,
Step 4: If $y = 0$, then pick another number $m$ and go to Step 2.
Step 5: Calculate digital signature $S = m^{-1} \ (H_2 + x \ H_1) \ mod \ n$ of an avatar (product),
Step 6: If $S = 0$, then pick another number $m$ and go to Step 2.

The digital signature of the avatar *S* provides for identifying users and making transactions on their behalf when creating a personalized tourism product. To implement a smart contract for the synthesis and authentication of the digital signature of an avatar (product), the stateful Ethereum blockchain technology is used. Authentication of digital avatars and tourism products is necessary to establish links between user travel profiles and selected tourist routes, and to detect fake avatars and invalid tourism products. Identified "bad" tourism products and compromised digital avatars are removed from the system and their hashes are added to the blacklist.

Figure 5 shows a structure of the information recommender system to build a tourist route.

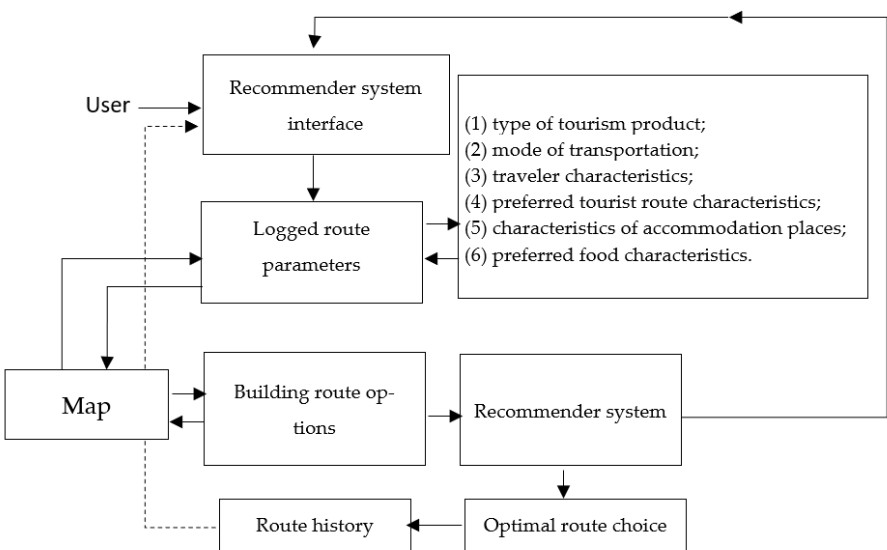

**Figure 5.** Structure of a tourist route information recommender system.

Since the system is recommendatory, it will offer several options according to preferences. If the user is not satisfied with any option, then he repeats the request and the neural network offers new options, excluding those that were rejected by him, and so on until there are no options left. If none of them fit anyway, then the user is invited to design the route himself by selecting points with sights on the map and manually select places for overnight stays and meals along the generated route. In the future, the manually generated route is added to the user profile, and it is determined which cluster the generated route corresponds to, the preferences of the user group for this cluster are set or a new cluster of preferences is formed, tied to the route, and then the set of route features and preferences is used to retrain the neural network for future recommendations to this type of user.

## 5. Discussion and Conclusions

In this paper, the authors have proposed a concept of universal information recommender system for building a personalized tourist route. The concept includes an original authorial DLT-based methodology for tourism product synthesis in order to consistently form a digital tourist avatar being a cyber-physical system. To form recommendations, it is proposed to cluster digital avatars of tourist profiles and tourism products for subsequent individualization of the tourism product offer.

Despite current dynamic research on tourism recommender systems, a lot of issues and challenges have yet to be resolved [21,49–52].

Tourism recommender systems are basically focused on interaction with the target user, thus emphasizing on the design of human-machine interaction within a cyber-physical system rather than on optimizing the function of predicting results. Therefore, when developing a RS, the expediency of using new forms of interaction, such as bots or speech interfaces, is updated.

Recently, the use of neural network and deep neural network methods [53–57] has become a key trend. Typically, deep learning algorithms use a sequence of non-linear processing layers to represent complex features [56]. However, while recognizing their effectiveness in some areas, for example, when creating sequences of recommendations with the implicit construction of an algorithm for solving a problem, it should still be recognized that this method does not allow solving all types of recommender problems.

To implement an individual tourist route, it is vital to make a decision either to visit the destination or to proceed the already started trajectory of attended POI [57,58]. In tourism, unlike, for example, e-commerce, there is no clearly defined catalog of recommended products, since preferences are often very individual and diverse [59–61]. Thus, it is important for a RS to help a tourist with any preferences to find an interested object based on personal cultural and mental characteristics when making a decision [62]. According to some authors [21,63,64], first, a RS should be able to "convince" a tourist of the correctness of its recommendations, especially if an object (POI) is not yet known to the tourist. Second, the recommended POIs should satisfy and provide the tourist with an unforgettable experience.

It should be emphasized that existing approaches often neglect a very important aspect, namely, contextual factors of the tourist route that shape travel experience: cultural characteristics of a tourist, weather conditions, travel accompaniment, travel period, etc.

As for online tourism market, its major players have not yet adopted these complex decisions and are currently offering recommendations far from being personalized, as they are either based on the regular user opinion or the popularity of tourist sites [63,64].

This case may be due to the difficulty of obtaining information about actual user behavior, that is, about actual experience of travelers and the performed sequence of actions [65–70]. Thus, a RS in tourism lacks data on user preferences to create effective and personalized recommendations [71].

In this article, the recommender system proposed by the authors solves this problem by creating a digital tourist avatar with actual information being collected and accumulated throughout the implementation of the entire tourist route. Based on this information, the construction and subsequent training of a multilayer fuzzy neural network are implemented. This provides an opportunity both to recommend a tourist objects that he liked in the past and to indicate new potential points of the tourist route based on the clustering of digital avatars of tourist profiles and tourism products. The importance of such aspects in the construction of personalized tourist routes is noted by many authors [72,73].

In this study, the authors have proposed the concept of representing a tourism product based on a distributed ledger technology (blockchain) that has great potential for creating intelligent tourism products and providing personalized services [74].

It is via the blockchain that trust is strengthened, and safer exchange of information, transparency and openness are ensured. It allows one to effectively "eliminate" intermediaries and provide the same powers to both suppliers and consumers of tourism services [75].

However, despite the promising benefits, the implementation of blockchain technology in the construction of intelligent tourist routes may face a number of problems. First, the conceptual complexity of blockchain technology and excessive market novelty create the biggest challenge for its practical implementation [76]. Moreover, some tourism organizations and tourism service providers are not ready to implement blockchain due to lack of experience and knowledge about this new breakthrough technology [77].

Other problems are related to security issues and lack of awareness about data security [78], the risks of losing personal data, keys, misplacement of tokens, and protecting the confidentiality of user personal records [79].

## 6. Future Research Directions

To implement the algorithm for developing the optimal tourism product, the authors plan to create and present an original method for clustering preferences of tourist pro-

files and tourism products and a method for comparative analysis (benchmarking) of tourism products.

Future research into the problem of developing a universal tourism recommender system is bound up with promising theoretical and methodological developments and system research in the field of designing a digital tourism cross-sectoral ecosystem based on public-private partnership, taking into account digitalization and integration of individual information systems of federal departments, regional authorities, and government services.

**Author Contributions:** Conceptualization, L.G.; formal analysis, L.G., A.F., M.K. and L.B.; writing—original draft preparation, L.G., A.F., M.K. and L.B.; collected data, L.G. and A.F.; review and editing, L.G. and A.F.; supervision, L.G.; writing—conceptualization, L.G.; methodology, L.G. and A.F.; project administration, L.G.; funding acquisition, L.G. All authors have read and agreed to the published version of the manuscript.

**Funding:** This research was funded by grant from the Russian Science Foundation (RSF) and Penza Oblast (Russia) (project No. 22-28-20524), https://rscf.ru/en/project/22-28-20524/.

**Data Availability Statement:** Not applicable.

**Acknowledgments:** Not applicable.

**Conflicts of Interest:** The authors declare no conflict of interest.

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
