# Peer review of "A Design Concept for a Tourism Recommender System for Regional Development"

_algorithms, doi:10.3390/a16010058_

Round 1
Reviewer 1 Report
The following issues should be taken into consideration to further improve the paper quality.
1. The abstract and conclusion need to be improved. The abstract must be a concise yet comprehensive reflection of what is in your paper. Please modify the abstract according to “motivation, description, results and conclusion” parts. The conclusion also need to be improved. I suggest extending the conclusions section to focus on the results you get, the method you propose, and their significance.
2. The introduction section needs to be rewritten with much better motivation and providing the context for this work. It should include:
(1). Contextualization
(2). Importance/Relevance of the theme
(3). Research question
(4). Objectives
(5). Structure of the paper
3. In Literature Review, the authors should analyze the shortcomings of other methods and highlight the advantages of their methods, rather than simply listing the related work.
4. Figure 4: The text should not exceed the frame.
5. The major contribution or focus of tourism information recommender system has been studied for long time and there are currently many related literatures which also study the same issue. Therefore, it is better to further expand the related work content by introducing more related papers published in recent two years such as the following: privacy-aware point-of-interest category recommendation in internet of things; interaction-enhanced and time-aware graph convolutional network for successive point-of-interest recommendation in traveling enterprises; a long short-term memory-based model for greenhouse climate prediction.
Author Response
We would like to thank the reviewer for your comments, we respond to each point below and we identify changes in the revised manuscript.
- Thank you for this suggestion. We modified and improved the abstract as follows. “Despite of tourism infrastructure and software, the development of tourism is hampered due to the lack of information support, which encapsulates various aspects of travel implementation. This paper highlights a demand for integrating various approaches and methods to develop a universal tourism information recommender system when building individual tourist routes. The study objective is proposing a concept of a universal information recommender system for building a personalized tourist route. The developed design concept for such a system involves a procedure for data collection and preparation for tourism product synthesis; a methodology for tourism product formation according to user preferences; the main stages of this methodology implementation. To collect and store information from real travelers, this paper proposes to use elements of blockchain technology in order to ensure information security. A model that specifies the key elements of a tourist route planning process is presented. This article can serve as a reference and knowledge base for digital business system analysts, system designers, and digital tourism business implementers for better digital business system design and implementation in the tourism sector”.
- Thank you for this suggestion (Kindly see Lines 64-66; 83-89; 101-102; 105-130).
- Thank you for this suggestion. We made all the necessary additions for improving this section to address the given recommendations (Kindly see Lines 132-136; 165-188; 216-227).
- Thank you for this suggestion. We improved the Figure.
- Thank you for this suggestion. We added in the reference list and cited them in the paper the following works: 1) Y. Liu et al., "Interaction-Enhanced and Time-Aware Graph Convolutional Network for Successive Point-of-Interest Recommendation in Traveling Enterprises," in IEEE Transactions on Industrial Informatics, vol. 19, no. 1, pp. 635-643, Jan. 2023, doi: 10.1109/TII.2022.3200067 2) L. Qi, Y. Liu, Y. Zhang, X. Xu, M. Bilal and H. Song, "Privacy-Aware Point-of-Interest Category Recommendation in Internet of Things," in IEEE Internet of Things Journal, vol. 9, no. 21, pp. 21398-21408, 1 Nov.1, 2022, doi: 10.1109/JIOT.2022.3181136.
Reviewer 2 Report
Summary:
This paper introduces a universal tourism information recommender system based on individual tourist routes. The topic is interesting, but the paper treats it superficially.
major comments:
-the topic is treated superficially, specifically the recommendation algorithm, which is supposed to be the most important part of the paper, did not give the details of how the recommendation is done, it just lists high-level generic steps, such as 'developing a quantitative matrix' 'identify preference matrix', 'cluster users' ...etc.
- the use of blockchain is not relevant at all, although it is true that blockchain can provide a secure way of storing and managing data, but the focus of this paper is on recommendation rather than data management.
minor comments:
- some references are cited without any explanation of their content and how they relate to the current research paper. For instance, in the second line of the related work section, [10-20] were cited without elaborating what is the content of the reference [10-15].
- the text in figure 1 is unreadable.
- quoting from the related work section 'But they are unable to provide feedback in the form of images that are geographically linked to certain places.'. This doesn't seem right, Google Maps has images of almost all tourist attractions places.
Author Response
We would like to thank the reviewer for your comments, we respond to each point below and we identify changes in the revised manuscript.
Major comments:
-the topic is treated superficially, specifically the recommendation algorithm, which is supposed to be the most important part of the paper, did not give the details of how the recommendation is done, it just lists high-level generic steps, such as 'developing a quantitative matrix' 'identify preference matrix', 'cluster users' ...etc.
Thank you for this suggestion. We tried to add some details about the recommendation algorithm. We added the process of selection of Optimization criteria depending on the mode of travel of the tourist; the description of the Multi-criteria method for the synthesis of tourist routes (Kindly see Lines 327-343; 375-381; 446-448)
- the use of blockchain is not relevant at all, although it is true that blockchain can provide a secure way of storing and managing data, but the focus of this paper is on recommendation rather than data management.
Thank you for this suggestion. The end result of the recommendation service is the synthesis of alternative TPs with the recommendation of the optimal route, followed by navigation support for the traveler to adjust the route, taking into account the current situation, associated costs, user wishes, as well as to save notes and reviews of the traveler about points on the route for accumulation -niya own experience, analysis and synthesis of other tourism products. Each product synthesized and mastered during the journey is stored in a hierarchical database synthesized in the form of a distributed ledger (blockchain) and hashed for unique identification in order to protect against unauthorized access and compromise. This is necessary, because in modern travel services, booking services, ratings, etc. predominantly positive reviews that are written by managers of tourist facilities, hired bloggers or bots. At the same time, negative reviews and descriptions are deleted, since access to them is determined by system administrators of information services of tourist sites. That is why, in order to collect and store information from real travelers, this paper proposes to use elements of blockchain technology to ensure information security. Kindly see Lines 105-119.
Minor comments:
- some references are cited without any explanation of their content and how they relate to the current research paper. For instance, in the second line of the related work section, [10-20] were cited without elaborating what is the content of the reference [10-15].
Thank you for this suggestion. We corrected the citing of the literature and added extra information about the content
- the text in figure 1 is unreadable.
Thank you for this suggestion. We corrected it.
- quoting from the related work section 'But they are unable to provide feedback in the form of images that are geographically linked to certain places.'. This doesn't seem right, Google Maps has images of almost all tourist attractions places.
Thank you for this suggestion. We corrected this sentence as follows: But they sometimes are unable to provide feedback in the form of images that are geo-graphically linked to certain places. Kindly see Line 214.
Reviewer 3 Report
The paper proposes a recommender system for guiding tourists through their trip based on multiple criteria. The authors present some related systems and compare them to their approach.
There are six criteria that need to be filled manually or automatically and then the process for designing the trip is completed in six steps.
Criteria are presented in some level of detail, but it is not clear enough how these are exploited towards designing a trip that fulfills the criteria. There are descriptions about how the TSP algorithm and blockchain can be employed, but these aid just some parts of the algorithm proposed. For example, for steps 2 and 4 in page 7, only some generic actions are described.
In order to improve the quality of their paper, authors are proposed to:
- Refer to the usage of TSP, blockchain, neural networks in other similar applications
- Describe in more detail how every parameter of the tourist product proposed is exploited in order to form the proposed trip
- Explain all stages of the algorithm and not just the parts related to TSP and blockchain
- Provide a case study with sample data to illustrate the data flow from the stage of setting the parameters of the trip until the final suggestion
- Describe what happens if the traveler does not agree with and alter some of the suggestions. Are the rest of them recalculated somehow?
Author Response
We would like to thank the reviewer for your comments, we respond to each point below and we identify changes in the revised manuscript.
1. Refer to the usage of TSP, blockchain, neural networks in other similar applications
Thank you for this suggestion. We made all the necessary additions for improving this section to address the given recommendations (Kindly see Lines 132-136; 165-188; 216-227).
2. Describe in more detail how every parameter of the tourist product proposed is exploited in order to form the proposed trip
Thank you for this suggestion. We tried to add some details about the recommendation algorithm. We added the process of selection of Optimization criteria depending on the mode of travel of the tourist; the description of the Multi-criteria method for the synthesis of tourist routes (Kindly see Lines 327-343; 375-381; 446-448)
3. Explain all stages of the algorithm and not just the parts related to TSP and blockchain
Thank you for this suggestion. Kindly see Lines 327-343; 375-381; 446-448. To implement the algorithm for developing the optimal tourism product, in future research the authors plan to create and present an original method for clustering preferences of tourist profiles and tourism products and a method for comparative analysis (benchmarking) of tourism products.
4. Provide a case study with sample data to illustrate the data flow from the stage of setting the parameters of the trip until the final suggestion
Thank you for this suggestion. In this research we propose a conceptual framework of the recommendation system. In the future research we’re panning to continue with a case study and test it on a real data.
5. Describe what happens if the traveler does not agree with and alter some of the suggestions. Are the rest of them recalculated somehow?
Since the system is advisory, it offers several options according to preferences. If the user is not satisfied with any option, then he repeats the request and the neural network offers new options, excluding those that were rejected by him. And so on until there are no options left. If none of them fit anyway, then the user is invited to design the route himself by selecting points with sights on the map and manually select places for overnight stays and meals along the generated route. In the future, the manually generated route is added to the user profile, it is determined which cluster the generated route corresponds to, the preferences of the user group for this cluster are set or a new cluster of preferences is formed, tied to the route, and then the set of route features and preferences is used to retrain the neural network for future recommendations to this type of user .
Round 2
Reviewer 2 Report
My previous comments have been addressed, the manuscript is in much better shape now. Additionally, you can improve related work section by covering the following related works:
[1] Hong, Minsung, and Jason J. Jung. "Multi-criteria tensor model for tourism recommender systems." Expert Systems with Applications 170 (2021): 114537.
[2] Dhelim, Sahraoui, Nyothiri Aung, Mohammed Amine Bouras, Huansheng Ning, and Erik Cambria. "A survey on personality-aware recommendation systems." Artificial Intelligence Review 55, no. 3 (2022): 2409-2454.
Author Response
Dear reviewer, thank you for this suggestion. We added these publications in the reference list and cited them within the text.
Reviewer 3 Report
In the revised version, the authors tried to enrich related work, they explained in more detail the role of TSP method in their proposal and they elaborated on the steps of the methodology. However, the methodology as described still seems to be a scattered exposition of steps that are not interlinked enough for someone to understand fully how the recommender system works from beginning to end. Some of these details, including a case study, are expected as future work as authors write. In my opinion though, they are necessary in the current version of the paper to improve understandability.
Author Response
Dear reviewer, thank you very much for the given comments. We tried to rework the Methodology section completely. We described in detail all the stages of the proposed algorithms. Kindly see lines 328-573. The suggestion about a case study is reasonable, meanwhile in this paper it is problematic for us to present it.